A mathematical model for predicting glucose levels in critically-ill patients: the PIGnOLI model

Zhang Zhongheng zh_zhang1984@hotmail.com
Department of Critical Care Medicine, Jinhua Municipal Central Hospital, Jinhua Hospital of Zhejiang University , Zhejiang , PR China
Wang Henry
Electronic publication date: 2015 Jun 9
Publication date: 2015
Volume: 3
Electronic Location ID: e1005
Received 2015 Apr 14; Accepted 2015 May 18
Copyright: © 2015 Zhang
Copyright year: 2015
Copyright holder: Zhang
License: This is an open access article distributed under the terms of the Creative Commons Attribution License, which permits unrestricted use, distribution, reproduction and adaptation in any medium and for any purpose provided that it is properly attributed. For attribution, the original author(s), title, publication source (PeerJ) and either DOI or URL of the article must be cited.
License URL: https://creativecommons.org/licenses/by/4.0/

Keywords: Big data, Intensive care unit, Insulin, Dosage, Glycemic control, Mathematical model

Funding: The author declares there was no funding for this work.

==============================
Background and Objectives. Glycemic control is of paramount importance in the intensive care unit. Presently, several BG control algorithms have been developed for clinical trials, but they are mostly based on experts’ opinion and consensus. There are no validated models predicting how glucose levels will change after initiating of insulin infusion in critically ill patients. The study aimed to develop an equation for initial insulin dose setting.

Methods. A large critical care database was employed for the study. Linear regression model fitting was employed. Retested blood glucose was used as the independent variable. Insulin rate was forced into the model. Multivariable fractional polynomials and interaction terms were used to explore the complex relationships among covariates. The overall fit of the model was examined by using residuals and adjusted R-squared values. Regression diagnostics were used to explore the influence of outliers on the model.

Main Results. A total of 6,487 ICU admissions requiring insulin pump therapy were identified. The dataset was randomly split into two subsets at 7 to 3 ratio. The initial model comprised fractional polynomials and interactions terms. However, this model was not stable by excluding several outliers. I fitted a simple linear model without interaction. The selected prediction model (Predicting Glucose Levels in ICU, PIGnOLI) included variables of initial blood glucose, insulin rate, PO volume, total parental nutrition, body mass index (BMI), lactate, congestive heart failure, renal failure, liver disease, time interval of BS recheck, dextrose rate. Insulin rate was significantly associated with blood glucose reduction (coefficient: −0.52, 95% CI [−1.03, −0.01]). The parsimonious model was well validated with the validation subset, with an adjusted R-squared value of 0.8259.

Conclusion. The study developed the PIGnOLI model for the initial insulin dose setting. Furthermore, experimental study is mandatory to examine whether adjustment of the insulin infusion rate based on PIGnOLI will benefit patients’ outcomes.

Introduction

Blood glucose (BG) control is of paramount importance in critically ill patients. A large body of evidence on BG control in intensive care unit (ICU) has emerged (Arabi et al., 2008; Griesdale et al., 2009; Van den Berghe et al., 2001) and has lead to elaboration of international guidelines (Ichai et al., 2009; Qaseem et al., 2011), which state that both hypoglycemia and hyperglycemia are associated adverse outcomes. However, these guidelines simply give a target of BG to achieve without elaborating on specific algorithms to achieve such a target range.

There are many algorithms on the dosing of insulin to control BG. In the well-known study by NICE-SUGAR Study Investigators et al. (2009), the specific protocol on the dosing of insulin was given, aiming to reach a steady BS within target ranges in both aims. This protocol categorized dosing strategies on whether insulin was first initiated or continued. In another study conducted in Australia, a locally developed protocol was found to be effective in maintaining BG in the target range (Breeding et al., 2014). However, several common features of these protocols include: (1) they were developed largely by expert opinion and experiences. These experts can be nurses, pharmacists, intensivists and investigators; (2) they only take into account a limited number of clinical variables such as the measured BG and the trend of BG changing after initiation of insulin pump. However, there are numerous factors that can influence insulin sensitivity. These factors include but are not limited to the history of diabetes, severity of illness, liver function and route of glucose intake.

It is important in critically ill patients to have the ability to predict response to medications. Regression modeling has been used to model predicted drug response. Medication dosing by using this approach is useful for drugs that have a narrow therapeutic window and require frequent dosing adjustment to reach a predefined target range. In a critical care setting, heparin dosing is a good example and has been investigated by using this regression modeling approach The objective of this study was to derive Predicting Glucose Levels in ICU (PIGnOLI), a mathematical model predicting the change in glucose level resulting from the initiation of insulin infusion in critically ill patients.

Methods

Design

The retrospective study encompassed analysis of a Multiparameter Intelligent Monitoring in Intensive Care II (MIMIC-II), a large clinical database of critically ill patients. Because the study utilized an open access clinical database, formal IRB approval was not required.

Data source

MIMC-II is a large registry of intensive care unit patients treated at Beth Israel Deaconess Medical Center, Boston, Massachusetts. Patient information on demographics, laboratory findings, imaging study, vital signs and progress notes was available (Saeed et al., 2011). MIMIC contains data on over 30,000 patients admitted during the period of 2001–2008. The database comprised varieties of ICUs, including the medical, surgical, coronary, and cardiac surgery recovery care units. ICU stays separated by less than 24 h were considered as one episode of ICU stay. Data were collected from electronic health data and all information produced during hospital stay were stored in the database.

All data were extracted by using structural query language (SQL) programming language from the database (Zhang, 2015). The Institutional Review Boards of the Massachusetts Institute of Technology (Cambridge, MA) and Beth Israel Deaconess Medical Center (Boston, MA) approved the establishment of the database. De-identification was performed to ensure patients’ confidentiality. Access to the database was approved after completion of the NIH web-based training course named “Protecting Human Research Participants” by the author Z.Z. (certification number: 1132877).

Selection of subjects

All adult patients were considered potentially eligible for this study. Patients actually received continuous insulin infusion were included. Children and neonates were excluded.

Outcomes

The primary outcome was retested BG level (mg/dl). The value, date and time of each BG were recorded in medical record. BG values can be those from blood chemistry and fingerstick, and the differentiation between venous glucose measurements and fingerstick levels was not performed in current analysis.

Clinical variables

Specific SQL programming languages for data extraction are shown in Supplemental Information 1. Comorbidities including diabetes, liver failure, congestive heart failure and renal failure were extracted because I felt that they may influence the sensitivity to insulin therapy. Laboratory parameters including bilirubin, C-reactive protein, serum creatinine and lactate were extracted. A total number of 1,117,076 BG measured with finger stick and in chemistry were extracted. A total number of 480,560 episodes of insulin rate were extracted.

Simultaneous use of intravenous (IV) total parental nutrition (TPN) and dextrose were extracted from the database. Different concentrations of dextrose were transformed to 5% dextrose (e.g., a volume of 10 ml 10% dextrose equals to 20 ml 5% dextrose). A total of 558,634 episodes of oral feeding (PO) containing glucose were extracted for its volume and time. All events were based upon charted time.

Data analysis

The objective of the analysis was to establish a linear regression equation between retested BG and insulin rate, controlling for other potential confounders. A data-driven approach means that the form of the equation was determined by data, depending on statistical significance. All variables thought to be associated with insulin sensitivity were extracted from the database and were considered for their inclusion in the model at outset.

I employed a multivariable fractional polynomial (MFP) method to construct the main effect model. The method combines backward elimination of statistically non-significant covariates with an iterative examination of the scale of continuous variables. MFP specifies two levels of significance levels: α1 = 0.15 for the test for exclusion and addition of variables to the equation and α2 = 0.05 to assess significance of fractional polynomial transforms of continuous variables. One degree of freedom was assigned to dichotomous variables and two-term fractional polynomials with 4 degrees of freedom were assigned to continuous variables. Continuous variables were modeled using closed test procedure, determining whether the covariate should be dropped from model at α1. Then, α2 = 0.05 was employed to test the need for transformation of the variable. The best two-term transformation was compared to the linear term by employing a closed test procedure. If the two term model is significantly better than the linear one at α2 = 0.05, the two term model is then compared to the one-term model. Otherwise, the linear term was retained in the model. Interactions were explored and terms with p < 0.05 were retained in the model.

The overall fit of the model was assessed by using R-squared, which is a reflection of the variance that can be explained by the model. Influential observations were evaluated by examining the leverage, Cook’s D and DFITS. Influential observations were excluded and the model was refitted by using MFPIGEN module. If the new model was significantly different from the original one, the original model would be reconsidered for the more parsimonious one. For example, some fractional transformation would be dropped and interaction terms could be dropped if the likelihood ratio test showed p > 0.05. R-squared values of the new parsimonious model would be compared to the original one to see whether the fitness was good enough. The whole dataset was split into two subsets: the training subset and the validation subset. Observed values of covariates were substituted into the fitted model to derive linear prediction. I then performed a regression model with linear prediction of the training subset as dependent variable and linear prediction of the validation subset as independent variable. The regression coefficient should be close to 1 and statistically significant at p < 0.05 if the model fits well to the validation subset.

All statistical analyses were performed by using Stata 13.1 (StataCorp, College Station, Texas, USA) and R software (R 3.1.1). Statistical significance was considered at p < 0.05.

Results

A total of 6,487 ICU admissions requiring insulin pump therapy were identified from the dataset. The dataset was randomly split into two subsets at 7 to 3 ratio. The training subset comprised 4,593 observations and the validation subset comprised 1,894 observations.

Model exploration and development

The results of initial model fitting are shown in Table 1. The continuous variables including glucose, interval, dextrose rate and insulin rate were FP transformed and there were significant interactions between insulin rate and two terms of glucose. Glucose was transformed by two-term FP with the power of −0.5 and 1. Interval was transformed by two-term FP with the power of −2 and 1. Dextrose rate was transformed by one term FP with the power of 0.5. There were two interaction terms between insulin rate and glucose because glucose was modeled with two terms. The overall fit of the model was thought to be good with an adjusted R-squared value of 0.8449.

Table 1 Multivariable linear regression model to predict retested blood glucose (mg/dl) after initiation of insulin infusion.

Covariatesa	Coefficient	Standard error	Lower limit of 95% CI	Upper limit of 95% CI	p	
Glucose−0.5	34.33	8.81	17.07	51.59	<0.001	
Glucose-1.96	94.00	1.60	90.85	97.14	<0.001	
(Insulin rate-2.85)	−1.06	0.33	−1.70	−0.42	<0.001	
(Time interval)2-8.13	−0.002	0.0006	−0.004	−0.001	<0.001	
Time interval-0.35	−18.90	2.30	−23.41	−14.39	<0.001	
(Dextrose rate)0.5	22.02	8.58	5.20	38.84	0.01	
PO volume	−0.02	0.01	−0.03	0.00	0.06	
TPN volume	0.07	0.03	0.02	0.12	0.01	
Lactate (mmol/l)	0.87	0.23	0.41	1.32	0.00	
History of congestive heart failure	2.64	1.18	0.33	4.95	0.03	
History of renal failure	−3.13	1.87	−6.78	0.53	0.09	
History of liver disease	4.50	2.03	0.52	8.47	0.03	
(Glucose−0.5) × (Insulin rate-2.85)	10.92	4.34	2.41	19.43	0.01	
(Glucose-1.96) × (Insulin rate-2.85)	1.60	0.67	0.28	2.91	0.02	
Constant	186.52	0.98	184.61	188.44	<0.001	
Notes.

Number of obs =4,593, F(14, 4578) = 1787.14, Prob >F = 0.0000, R-squared = 0.8453, Adj R-squared = 0.8449, Root MSE = 30.569.

PO by mouth, orally (from the Latin “per os,” by mouth)

TPN total parental nutrition

a Some covariates were centered and transformed with fractional polynomials.

Influential observations were examined by using regression diagnostics (Supplemental Information 2). By excluding these influential observations, I refitted the model and found that FP terms and coefficients were remarkably changed (Table 2). Glucose was transformed by two-term FP with the power of −2 and 1. Interval was transformed by two-term power of 3 and 3. The results showed that the model was not stable, most likely due to complexity of the FP assignment and multiple testing during model fitting. The FP terms were influenced by several influential observations.

Table 2 Refitting the regression model after excluding influential observations.

Covariatesa	Coefficient	Standard error	Lower limit of 95% CI	Upper limit of 95% CI	p	
(Glucose/100)−2	10.865	2.589	5.789	15.941	<0.001	
(Glucose/100)-1.96	92.193	0.985	90.261	94.125	<0.001	
Insulin rate-2.85	−0.861	0.316	−1.481	−0.241	0.007	
(Interval/100)3	−5.530	7.977	−21.168	10.109	0.488	
(Interval/100)3 × ln(interval/100)	77.272	19.426	39.189	115.356	<0.001	
[(Dextrose rate + 0.01)/100]0.5 − 9.3 × 10−6	5.584	1.703	2.247	8.922	0.001	
[(Dextrose rate + 0.01)/100]3 ×ln[(Dextrose rate + 0.01)/100] + 3.6 × 10−5	−4.011	1.106	−6.180	−1.843	<0.001	
PO volume	−0.013	0.008	−0.029	0.002	0.094	
TPN volume	0.091	0.030	0.031	0.150	0.003	
Lactate (mmol/l)	0.865	0.227	0.421	1.310	<0.001	
Congestive heart failure	2.376	1.151	0.120	4.632	0.039	
Renal failure	−2.983	1.817	−6.546	0.580	0.101	
Liver disease	4.254	1.977	0.379	8.129	0.031	
(Glucose/100)−2 × (insulin rate-2.85)	0.489	1.163	−1.792	2.769	0.674	
[(Glucose/100)-1.96] × (insulin rate-2.85)	0.189	0.358	−0.513	0.891	0.598	
Constant	185.875	0.910	184.091	187.659	<0.001	
Notes.

Number of obs = 4585, F(15, 4569) = 1760.88, Prob >F = 0.0000, R-squared = 0.88525, Adj R-squared = 0.8520, Root MSE = 29.784.

PO by mouth, orally (from the Latin “per os,” by mouth)

TPN total parental nutrition

a Some covariates were centered and transformed with fractional polynomials.

The parsimonious model was fitted to address the problem of instability. Graphical presentation showed that although the interaction term was statistically significant, the magnitude was of marginal clinical significance (Fig. 3 in Supplemental Information 2). Therefore, I opted not to incorporate interaction terms in the parsimonious model. Figure 1 shows the scatter points predicted by FP model and simple linear model, and the two lines were close to each other. Visual inspection of the graph indicates the use of parsimonious model would not compromise the prediction accuracy of the model.

Figure 1 Graphical presentation of the BG predicted by the model including FP terms (red line) and the model with linear terms (blue line).

Both models appeared similar in predicting BG. The initial BG was controlled at its mean value of 195.9 mg/dl.

Final model and model validation

The final Predicting Glucose Levels in ICU (PIGnOLI) model is shown in Table 3. Insulin rate was significantly associated with blood glucose reduction (coefficient: −0.52, 95% CI [−1.03, −0.01]). Initial blood glucose was the most important determinant of retested blood glucose (coefficient: 0.89, 95% CI [0.88, 0.90]). Oral intake, TPN and dextrose infusion were all associated with blood glucose control. Furthermore, serum lactate and BMI were positively associated with retested blood glucose. The time interval was negatively associated with retested blood glucose level (coefficient: −0.18; 95% CI [−0.22, −0.14]). The PIGnOLI model showed an adjusted R-squared value of 0.84, which was not significantly different from the FP model with interaction terms (R-squared = 0.84). The PIGnOLI model was tested in the validation subset and the result showed that the coefficient between estimated retest glucose and observed retest glucose was 0.99 (95% CI [0.97–1.01]; p < 0.001). The adjusted R-squared value was 0.8259, suggesting that the model was well calibrated with the validation subset.

Table 3 Parsimonious model with linear terms and no interaction.

Covariates	Coefficient	Standard error	Lower limit of 95% CI	Upper limit of 95% CI	p	
Insulin rate	−0.52	0.26	−1.03	−0.01	0.05	
Glucose	0.89	0.01	0.88	0.90	<0.001	
PO volume	−0.02	0.01	−0.03	−0.00	0.05	
TPN volume	0.07	0.03	0.01	0.12	0.01	
BMI	0.10	0.06	−0.02	0.22	0.09	
Lactate (mmol/l)	0.95	0.23	0.50	1.41	<0.001	
Congestive heart failure	2.58	1.19	0.26	4.91	0.03	
Renal failure	−3.09	1.87	−6.76	0.58	0.10	
Liver disease	4.13	2.03	0.14	8.12	0.04	
Interval	−0.18	0.02	−0.22	−0.14	<0.001	
Dextrose rate (5%)	0.01	0.01	−0.01	0.03	0.20	
Constant	17.18	2.23	12.82	21.55	<0.001	
Notes.

Number of obs = 4,593, F(11, 4581) = 2251.71, Prob >F = 0.0000, R-squared = 0.8439, Adj R-squared = 0.8435, Root MSE = 30.698

PO by mouth, orally (from the Latin “per os,” by mouth)

TPN total parental nutrition; BMI: body mass index

Discussion

This study developed the PIGnOLI model for BG control in critically ill patients. A data-driven approach could be applied in this study because there is a large volume of retrospective data available for analysis. The widespread uses of electronic medical record systems have made this strategy possible. The present study provides a framework for predicting and modeling BG response. This approach may be useful for predicting medication response in this and other disease states.

Although there is a large body of evidence suggesting the importance of BG control in the intensive care unit (ICU), there is no empirical data on how to control BG (Fahy, Sheehy & Coursin, 2009). Several BG control algorithms have been developed for clinical trials, but they are mostly based on experts’ opinion and consensus. As a result, many patients assigned to a specific BG range cannot reach that range, or many times insulin rate adjustment are required before an optimal target is reached. Furthermore, substantial number of patients experience under- or over-control of BG because of insulin misuse and/or other disease-related factors. It is optimal in clinical practice that BG be accurately controlled within a short period of time. In the present study, I developed an equation for insulin adjustment, by considering comorbidities, laboratory findings and demographics. Glucose intakes such as TPN, dextrose infusion and PO intake during the analysis time were all considered. In critical care practice, these information need to be collected and put into calculators to estimate the initial rate of insulin infusion. Since these variables are routinely recorded in most ICUs, this could be done with just any electronic health record (HER) system.

Glycemic control in the present clinical practice is not based on data-driven approach. For example, in the well-known study by NICE-SUGAR Study Investigators et al. (2009), insulin dosing algorithm was based on whether insulin was first initiated or continued. The insulin rate was determined on the value of BG, taking previous BG into consideration. This protocol did not take into account of other variables such as concomitant dextrose infusion, baseline renal and liver functions. In another study conducted in Australia, a locally developed protocol was found to be effective in maintaining BG in target range (Breeding et al., 2014). The insulin rate was set according to the amount of BG fall, without considering other potential influential factors.

The predictors in the PIGnOLI model have biologic and clinical plausibility. For example, congestive heart failure was positively associated with blood glucose. In a cohort of 3,748 nondiabetic participants aged ≥65 years, Guglin and coworkers (2014) found that baseline heart failure was associated with subsequent development of diabetes mellitus within 3–4 years. Liver disease may contribute to hyperglycemia via insulin resistance and increase hepatic BG output (DeFronzo, 1988; Mitrakou et al., 1990). With respect to the association of renal failure with glycemic control, although the present study failed to found a significant association at p = 0.05, I still incorporated this factor into my model because renal function has been identified to be tightly related to BG levels (DeFronzo, Davidson & Del Prato, 2012). Serum lactate is a biomarker of tissue perfusion, and it increases markedly with hypoperfusion and hypoxia. My previous work has demonstrated that lactate is a strong predictor of clinical outcome in critically ill patients (Zhang et al., 2014; Zhang & Xu, 2014; Zhang, Xu & Chen, 2014). I propose that, since lactate is biomarker of circulatory shock, it is also a biomarker of stress response during severe illness. Stress response is a well-established contributor to insulin resistance and observed hyperglycemia (Santos, 2013).

Many drugs require careful dosing because their therapeutic and toxic doses are close to each other. Insulin is one such drug with a therapeutic dose that varies substantially across individual patients. More importantly, inappropriate dosing may cause catastrophic consequences such as infection, permanent neurologic defect and coma. Therefore, close monitoring of BG and frequent adjustment of insulin dose are mandatory. Due to the complexity of the PIGnOLI model, I programmed the PIGnOLI model in Excel format (Supplemental Information 3) to ease its use in clinical practice (Fig. 2). The users can input required variables and predict retested BG after a predefined time interval (<120 min).

Figure 2 A snapshot of the calculator for setting initial dose of insulin.

Several limitations need to be acknowledged in this study. The study was restricted to dosing at the initiation of the insulin pump and subsequent adjustment was not addressed; the difficulty lay in the complexity of data preprocessing. In a future study I will try to resolve these technical difficulties and provide further algorithms on how to adjust the insulin dose by incorporating initial response to insulin therapy in addition to covariates as reported in the present study. This analysis included only patients receiving insulin via pump and would not necessarily be generalizable to patients receiving insulin by some other route. However, the insulin pump is the most attractive mean to give insulin for critically ill patients, mostly due to its accuracy in dosing and the short-acting property. This study may suffer from the problem of multiple testing and model overfitting. This happened in my first model, in which several FP terms and complex interactions were incorporated. However, this model was found to be unstable by excluding several outliers. Therefore, I opted to employ simple linear terms and clinical irrelevant interactions were excluded. The PIGnOLI model was validated in split subset and was well fitted to the independent subset. One important limitation of the study is that the database is limited to one center, and thus the extrapolation of the PIGnOLI model to other systems and countries requires further validation.

In conclusion, the study developed the PIGnOLI model for the initial insulin dose setting. It may be favorable if this algorithm can be used in clinical setting for accurate BG control for critically ill patients. Furthermore, an experimental study is mandatory in order to examine whether insulin adjustment based on the PIGnOLI model will benefit patients’ outcomes. Before the PIGnOLI model can be used in clinical practice, it is also mandatory to compare the episodes of hypoglycemia and duration of hyperglycemia between groups using and without using the PIGnOLI model.

Supplemental Information

Supplemental Information 1 Details of data extraction by using SQL language

Click here for additional data file.

Supplemental Information 2 Supplemental digital content

Click here for additional data file.

Supplemental Information 3 Excel based Blood glucose control calculator in the intensive care unit

Click here for additional data file.

Additional Information and Declarations

Competing Interests

Author Contributions

The author declares there are no competing interests.

Zhongheng Zhang conceived and designed the experiments, performed the experiments, analyzed the data, contributed reagents/materials/analysis tools, wrote the paper, prepared figures and/or tables, reviewed drafts of the paper.

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
