# Peer review of "A mathematical model for predicting glucose levels in critically-ill patients: the PIGnOLI model"

_PeerJ, doi:10.7717/peerj.1005_

## Round 0.1 · original submission · Major Revisions

Thank you for your very interesting manuscript. You present a timely and fascinating approach to glycemic control in the ICU, and I hope that one day your method will gain traction in clinical practice. I am requesting major revisions to hone your message and enhance the readability of your paper. I would anticipate one more round of edits after you address these concerns.

-Reviewer 1 was very confused about the importance of the work and found the work far too technical. Can you revise the paper to: 1) better address the importance of the work and 2) explain the methodology in far simpler terms? (You do not need to provide a point-by-point response to Reviewer 1 comments.)

-Reviewer 2 has a few specific comments that you should address. (Please provide a point-by-point response to each comment.)


In addition, please respond point-by-point to my comments below:

-Title: How about changing to “A Mathematical Model for Predicting Glucose Levels in Critically-Ill Patients”

-Throughout the paper you talk about a “data driven approach.” I recommend minimizing the use of this term as it is a bit distracting.

-The first paragraph of the Discussion is excellent and well describes the issue at hand in motivating your analysis. Can you bring some of this verbiage to the introduction and abstract?

-Abstract; “There is no empirical data on how to set initial insulin dose for glycemic control.” How about changing here (and elsewhere) to “There are no validated models predicting how glucose levels will change after initiating of insulin infusion in critically ill patients”?

-Abstract; The final model is missing! While you do not want to include the regression coefficients, at least list something like; “The selected prediction model included the variables …..”

-Introduction para 1 – Try to provide a more basic layperson’s introduction to glycemic control in the critically ill.

-Introduction; Your objective is not well stated. Consider; “The objective of this study was to derive a mathematical model to predicting the change in glucose level resulting from the initiation of insulin infusion in critically ill patients” Try to revise the second paragraph to better and more concisely state your objective.

-Methods – Please revise the methods section using subsections like; Design, Setting, Data Source, Selection of Subjects, Outcomes, Clinical Variables, Data Analysis. You are missing a lot of important methodological information. By following these framework, you will have a more detailed methods section.

-Methods – Please provide a lot more information about MIMIC-III. What is MIMIC? Which hospitals contribute to mimic? What patients are included? How did MIMIC obtain patient data? How were variables defined?

-Methods – I recommend not using the term “big data.” Rather, consider; “I used data from MIMIC-III, a large international multicenter registry of 30,000 intensive care unit patients.”

-Methods Data Extraction - this paragraph needs revision. It is not clear how MIMIC obtained and organized glycemic data on each patient and how you organized these data for the analysis.

-Methods Statistical Analyss – Revise line 1 to something like; “The objective of the analysis was….”

-Results – What is the final model? This should be the main feature of your paper, yet I had to dig to find it.

-Results – Paragraphs 2 and 3 are far too detailed. Please try to explain your model selection process in much simpler terms. Para 2 and 3 should be reduced to a single concise paragraph.

-Discussion – can you add a paragraph summarizing the current international approaches to glycemic control, detailing how they are not data-driven? Has anyone else tried to derive a glycemic response model?

-Discussion para 2 line 169 – This paragraph is arguably not needed. If you want to preserve it, reframe the paragraph; “The predictors in the selected model have biologic and clinical plausibility. For example...”

-Can you develop a name/acronym for your model? This will make referring to “the model” a lot easier. “Predicting Glucose Levels in ICU”, “Blood Glucose Control in the ICU” or something like this. http://acronymcreator.net/ can help you to choose an acronym.

-Discussion – What are future needs/studies before we implement the model in practice?

-Discussion – Please comment on the complexity of the model how that might impact clinical practice. –i.e. if widely embraced, users will need a calculator to operate the model.

-Tables 1, 2 and 3 – Combine these into a single table. Delete the SE column. List variable names in plain words (“Glucose” – not “Igluc_1”)

-Drop Figures 1, 2 and 3 or move them to supplemental materials – they are not needed.

-Fig 4 – VERY interesting. You should feature this in the Discussion. Here and elsewhere, the model determines “predicted glucose response to insulin infusion,” not “Blood glucose control in the ICU.” Consider some changes to the title.

Reviewer 1 ·

Basic reporting

See "General Comments"

Experimental design

See "General Comments"

Validity of the findings

See "General Comments"

Additional comments

The main issue with this paper is that it is much too complicated for the average intensivist. There are multiple complex algorithms used to derive what is supposed to be a simple, easy-to-use method to figure out the best initial dose of insulin using an infusion. Although it is important to derive how you got to your conclusion, there is far too much complex mathematics for the average reader to understand, leaving them confused on the actual aim of the paper.
The next issue is simply put: Where is the actual equation they continue to refer to? Figure 4 is supposed to address this and what it appears to be is a screen shot from a computer or smart phone application that the user can input values. This should be made clear in the beginning of the paper, however it isn’t so the reader is left guessing (at least this reader was).
The remainder of the figures are very hard to interpret and add little to the whole of the paper.
This reviewer was also somewhat disappointed in the apparent lack of medicine mentioned in this paper. As a practicing intensivist I am looking to read articles that are going to add to my practice. This article leaves me scratching my head looking for the big picture.
Overall I feel this article is not suited for a medical journal. It would maybe be more appropriate to place this article in a statistics or medical mathematics journal. The more useful paper would be using this equation/application for a prospective experiment and use this as the experimental arm and using the standard approach as the control.

·

Basic reporting

Thank you for the opportunity to review this manuscript by Zhang, which sets out to develop a model capable of predicting re-test blood glucose levels following insulin administration in the ICU, an important and under-studied aspect of care for critically-ill patients. The paper is generally well-written and easy to follow. The arguments made for the utility of the presented algorithms are strong, with reference to prior work on heparin dosing. It is clear that using such an approach in the ICU could provide healthcare providers with valuable information related to how patients may respond. I would recommend further editing throughout for typographical errors.

Experimental design

Overall, I believe the methods are thoroughly described, something on which the author should be commended. I did note several issues which should be addressed.

1) Throughout the manuscript and even in the title, the author refers to blood glucose control. However, this is not what is presented in the models provided. The author gives us a method for predicting BG re-test levels, not necessarily control. I think an additional analysis focused on achieving some level of control would be helpful. Specifically, it would be helpful to see if the employed data driven approach could be used to determine this as a dichotomous variable with relevant established cutoffs. Then, it would be useful to examine model discrimination with respect to whether or not the patient will achieve control. This analysis would correspond more directly with the title of the manuscript and objectives of the analysis.

2) The author presents several different models for re-test BG, one with fractional polynomial terms for continuous variables and one with linear terms. The model including only linear terms is easier to interpret and implement. However, I feel that this as well as the fact that the FP terms changed after excluding outliers is not necessarily a reason to abandon the MFP selected model, especially because the model selection process did suggest that these terms offered some improvement over the linear terms based on tests included in the closed-test procedure. Further justification for this choice, beyond the fact that exclusion of outliers/influential points changed the results, is warranted. Importantly, if this model were to be used in clinical practice, there would likely be outliers and some complexity that could not be easily removed or handled.

Validity of the findings

The author used split-sample validation to assess internal validity and provided a thorough outlier/leverage analysis that adds to the robustness of the findings. The model selection and evaluation process are clearly described and seem appropriate. The dataset employed in the analysis is publically available, and codes used for extraction of relevant data are provided. I do note several issues that if resolved, could help to clarify some of the results.

1) Examining the tables, it would be helpful for the p-values to be reported consistently throughout. For instance, table 3 should be taken to the same number of digits as presented in table 2. This would facilitate comparison of the terms in each model.

2) It would be helpful to see a plot of the actual BG re-test values and that predicted by both the model including FP terms and the model with linear terms. Graphical inspection could lend support to the contention that the model with linear terms was sufficient.

3) This analysis included only patients receiving insulin via pump and would not necessarily be generalizable to patients receiving insulin by some other route. This should be clarified more prominently in the limitations section of the discussion.

Additional comments

This work by Zhang is generally a well-written with clear and appropriate methods. I believe the work could be strengthened substantially with several additions: 1) a formal prediction of BG control as opposed to BG re-test measure; 2) more scrutiny on the choice between the MFP selected model and the model with simple linear terms; 3) plot of the raw BG re-test data with the predictions from both models (tables 2 and 3) for comparison.

---

## Round 0.2 · Minor Revisions

Thank you for your continued work on this very interesting paper.

I made suggested edits to your Word file, as well as some comments. I append it as a PDF If you need the original Word doc please email the Editorial Office ([email protected]).

Please also address the reviewer's comments.

Thanks.

·

Basic reporting

Thank you for the opportunity to review this re-submission by Zhang. Overall, I feel that the manuscript has been strengthened through revisions and clarifications offered by the author.

Minor Comments:
-Please review the submitted tables. In order for these to be standalone, units need to be included and covariate names/acronyms should be clarified.

Experimental design

Minor Comments:
-In the methods section on page four, more description of the setting is needed. Needs to include the country, region, and study years.
-Instead of saying "Critical care big data was utilized for the present study" maybe replace with "An expansive critical care database was used for the present study."
-Revise "Selection of Subjects," as the analysis was limited to those actually receiving insulin, not those who needed it. These may not necessarily be the same group.
-Expand "Outcomes" section. Include information here on how BG was measured.

Validity of the findings

Minor Comments:
-In limitations please make note that this was a single system study in the US. Provide your interpretation of whether your results would or would not be generalizable to other systems and other countries.
-Consider extending your discussion of what steps need to be taken for the model to be incorporated into critical care practice. Could this be done with just any EHR system or is something specialized required? Perhaps this work (and the database you used) could serve as a blueprint for what kind of data items need to be collected regularly for implementation to be successful.

---

## Round 0.3 · accepted · Accept

Thank you for your revisions. I look forward to seeing the paper in print.